# MSTFORMER: MULTISCALE SPATIOTEMPORAL MOTION-AWARE TRANSFORMER NETWORK FOR EFFECTIVE AI-GENERATED VIDEO DETECTION

## ABSTRACT

Recent AI-generated videos (*e.g.*, Veo3) are growing increasingly realistic and indistinguishable from real videos. Current existing detectors usually rely on artifacts present in earlier or inferior generations, resulting in poor generalization to the newly published generators. To address the challenge of newly generated videos, we propose a novel dataset, AIDetection, for the AI-generated video detection task. The proposed AIDetection dataset contains 39,298 real and 19,731 generated videos from 27 diverse sources, specifically designed to evaluate cross-generator generalization under out-of-distribution settings. For the real videos, the motion of moving objects and the background show clear distinctions. Based on this observation, in this paper, we introduce a novel Multiscale Spatiotemporal motion-aware modeling Transformer framework (MSTformer) for the AI-generated video detection task, which learns motion-aware discriminative representations from both local and global viewpoints. Specifically, a novel motion-aware spatiotemporal downsampling mechanism is designed to capture local motion discrepancies between real and generated videos. Further, to prevent the discriminative cues from being weakened, we also employ a cross-scale semantic contrastive learning mechanism implemented on multiscale spatiotemporal features, enabling the model to maintain the global discriminative ability. Extensive experiments on three benchmark datasets (*i.e.* AIDetection, GVF, and GenVideo) demonstrate that MSTformer achieves superior cross-domain generalization performance, especially on the OOD setting.

## 1 INTRODUCTION

At present, the content of AI-generated videos from Emu (Girdhar et al., 2023) to Veo3 (DeepMind, 2025) is becoming increasingly high-quality, especially with advancements in diffusion technology, *e.g.*, Stable Video Diffusion (Ho et al., 2022b). How to correctly distinguish whether a video is AI-generated or real plays an important role in numerous fields, for example, protecting the intellectual property rights and copyrights of videos, artificial intelligence security, and so on.

AI-generated videos usually exhibit coherent actions, natural camera movements, and convincing physical effects (Sun et al., 2025), which bring us huge challenges. In addition, the AI generation technology tends to constantly evolve, and detectable artifacts are rapidly diminishing. Existing detectors primarily rely on visual artifacts from low-quality generations, which fail to generalize to high-quality videos. In the meantime, most advanced commercial generators remain closed-source, which poses a critical challenge: how can we design a detector that learns intrinsic differences between AI-generated and real videos, while ensuring robust generalization across unseen generators and even more advanced ones in the future?

To address this challenge, we deeply investigate and explore the motion information as the discriminative clue. It is well known that motion dynamics is a fundamental element of videos. As shown in Fig. 1, by analyzing the dense displacement fields across multiple generators, we observe that generated videos exhibit motion patterns inconsistent with the physical world. Specifically, objects in generated videos often share correlated motion with backgrounds or unrelated regions, likely due to temporal dependencies introduced during generation. In contrast, real videos exhibit localized and

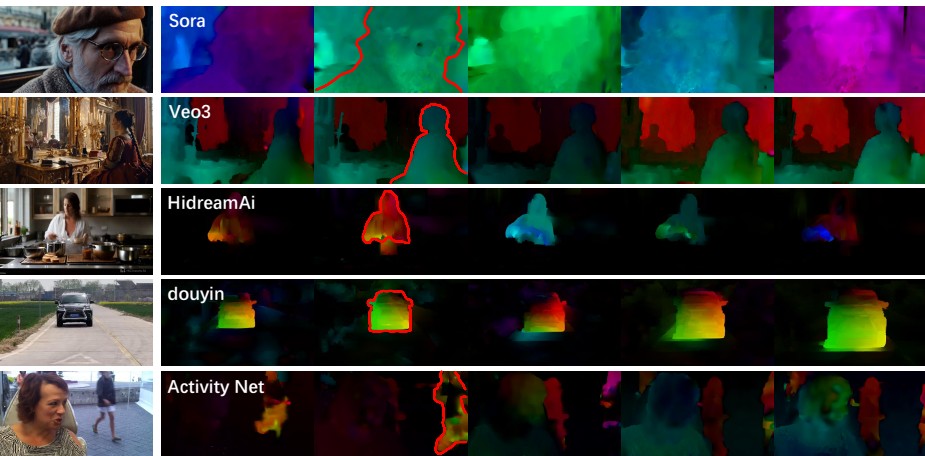

Figure 1: Visualization of Lucas–Kanade (LK) optical flow. Blue, yellow, cyan, and red indicate upward, downward, leftward, and rightward motion directions, respectively. Brighter colors correspond to higher motion speeds. The active subject in the current frame is outlined in red.

diverse motion, along with clear distinctions between foreground and background dynamics. Similar observations can also be also found in action recognition tasks that the motion being performed can evolve much faster than their subject identities, such as clapping, waving, shaking, walking, or jumping (Feichtenhofer et al., 2019). Observing from the spatial perspective, low-quality generations may reveal obvious artifacts, but high-quality videos are semantically indistinguishable from real ones. For the AI-generated videos, there are significant differences about texture and style variations with different video generators. For the real videos, diverse source material, resolution and editing types will bring inconsistent styles. Therefore, only make detection based on spatial features alone will be not enough. These observations motivate us incorporating temporal motion cues into effective AI-generated video detection.

To capture the underlying dynamic differences between real and generated videos, in this paper, we propose a novel Multiscale Spatiotemporal motion-aware modeling Transformer framework (MSTformer) for the AI-generated video detection task, which integrates both local and global motion-aware discriminative representation learning into a unified framework. To accurately capture local motion variance between the moving object and background, a novel motion-aware spatiotemporal downsampling mechanism is proposed to learn local motion discrepancies between real and generated videos. Motivated by the video domain generalization mechanism, we also design a cross-scale semantic contrastive learning module to enforce consistency across features extracted at different scales. The contrastive learning can help the model maintaining the global discriminative ability with the assistance of motion-aware information. In addition, we construct a dedicated dataset, AIDetection, which contains 19,731 generated videos from 24 sources and 19,298 real videos from three sources. The main advantage of AIDetection lies in that we introduce unknown or closed-source video generators to extend the verification of detection generalization. Experiments under out-of-distribution (OOD) settings on AIDetection validate the effectiveness in addressing the generalization limitations of existing detectors. The main contributions are summarized as follows:

- We propose a novel Multiscale Spatiotemporal motion-aware modeling Transformer framework (MSTformer) for the AI-generated video detection task, which integrates both local and global motion-aware discriminative representation learning into a unified manner. The fundamental differences in temporal motion relationships between generated and real videos can assist us in detecting AI-generated videos well.

- A novel motion-aware spatiotemporal downsampling mechanism implemented upon the temporal sequence is introduced to efficiently capture motion discrepancies between generated and real videos in the multiscale feature space. And a novel Cross-scale Semantic Contrastive Learning (CSCL) module is designed to enrich the diversity of the samples and guide the model learning more clear boundaries and discriminative features.

- To address the challenge of generalization on unknown or closed-source video generators, we construct a new AIDetection dataset, specifically for AI-generated video detection, consisting of generated videos from 24 different generators and real videos from three distinct sources, allowing comprehensive evaluation across multiple OOD testing scenarios.

## 2 RELATED WORK

### 2.1 VIDEO GENERATION METHODS

Video generation can be separated into two types, *i.e.*, text-to-video (T2V) and image-to-video (I2V). I2V aims to generate videos from a static image or an image with auxiliary instructions, and the goal is to produce plausible motion from initially static content while preserving spatial detail and temporal continuity. T2V directly generates videos from natural language text, which requires not only visualizing high-level semantic information but also adhering to fine-grained textual constraints. Prior video generation methods mainly adopt Generative Adversarial Networks (GANs) (Vondrick et al., 2016) (Aldausari et al., 2022) and autoregressive transformers (Xiong et al., 2024) *e.g.*, Video Pixel Networks (Kalchbrenner et al., 2017). Video Diffusion Models (VDM) (Ho et al., 2022b) introduced diffusion models to video generation and produce videos with significantly higher clarity and temporal consistency compared to GAN-based approaches. Later, Imagen Video (Ho et al., 2022a) leveraged the powerful pretrained text-to-image generator Imagen and adopted a cascade diffusion strategy, showing remarkable capability in high-resolution T2V synthesis.

### 2.2 GENERATED VIDEO DETECTION METHODS

Current research on AI-generated video detection mainly falls into four categories:

**(1) Multibranch spatiotemporal networks:** A representative algorithm of this line is the two-stream convolutional networks (Simonyan & Zisserman, 2014), followed by numerous extensions such as TSN (Wang et al., 2016) and I3D (Carreira & Zisserman, 2017). Bai et al. proposed AIGVDet, which captures abnormal textures and artifacts in low-quality generated videos. Ji et al. (2024) proposed DuB3D that jointly models appearance information from video frames and temporal dynamics across frames. Chang et al. (2024) designed a three-branch expert ensemble model based on raw frames, optical flow, and depth information.

**(2) Spatiotemporal consistency modeling:** Ma et al. (2024a) reduced AI-generated video detection to a two-dimensional problem and proposed DeCoF to disentangle spatial and temporal representations by mapping frames into a shared feature space. Chen et al. (2024b) introduced DeMamba, a plug-and-play module designed to enhance the detectors by identifying AI-generated videos through the analysis of inconsistencies in temporal and spatial dimensions. He et al. (2024) focused on both local and global temporal defects in generated videos. Kundu et al. (2025) designed UNITE, a universal network capable of detecting tampered and synthesized videos across diverse scenarios.

**(3) Large Multi-Modal Models:** MM-Det (Song et al., 2024) constructed a MMFR to detect forgery traces across different diffusion-generated videos. BusterX (Wen et al., 2025) treated AI-generated video detection as a visual reasoning task, which is built on the pretrained Qwen2.5-VL-7B (Razavi et al., 2019).

**(4) Methods with Diffusion Reconstruction Error:** Liu et al. (2024) adapted approaches from AI-generated image forensics and proposed DIVID, a video detection algorithm based on Diffusion Reconstruction Error (DIRE) (Wang et al., 2023). Vahdati et al. (2024) suggests that video generators leave unique traces that image-level detectors cannot capture.

However, a practical challenge for AI-generated video detection is that most high-quality generators are proprietary and closed-source, leaving researchers with limited knowledge of their internal design. As our focus is on improving generalization in detecting videos from unseen generators, we restrict our review to the essential background and do not delve further into generation methods.

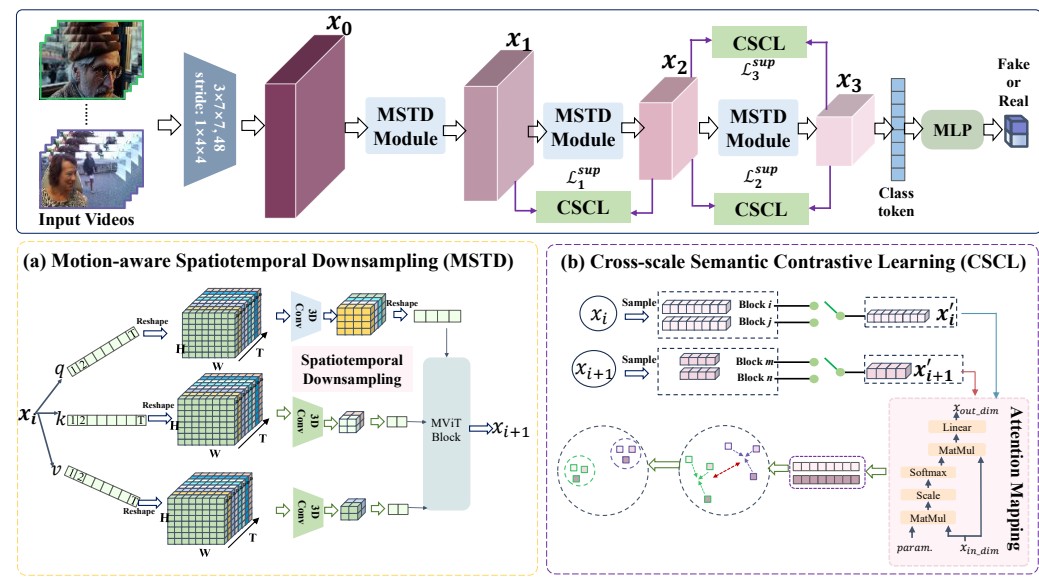

Figure 2: Overall pipeline of our proposed MSTformer method.

## 3 METHOD-MSTFORMER

### 3.1 FRAMEWORK OVERVIEW

As shown in Fig. 2, the proposed Multiscale Spatiotemporal motion-aware modeling Transformer framework (MSTformer) contains two core modules, *i.e.*, a Motion-aware SpatioTemporal Downsampling (MSTD) module and a Cross-scale Semantic Contrastive Learning (CSCL) module. MST-former adopts the MViTv2 (Li et al., 2022) as the backbone and it contains multiple stages with different resolutions. While we start with high-resolution inputs, the model progressively abstracts attention features through multiscale spatiotemporal downsampling. While we input a video, the input video will be mapped into four distinct scales, forming a multiscale feature pyramid, *i.e.*, $\{\mathbf{x}_0, \mathbf{x}_1, \mathbf{x}_2, \mathbf{x}_3\}$. Then, the MSTD module is implemented at the end of each stage, and CSCL mechanisms is implemented on different stages.

### 3.2 MOTION-AWARE SPATIOTEMPORAL DOWNSAMPLING MODULE

Prior work in video classification has explored and verified the important role of spatial or temporal features in constructing useful representations. In real videos, object motion dynamics and background dynamics should have clear distinctions. Analogously, the motion cues of AI-generated videos should also meet the requirements. Motivated by this observation, we integrate the motion dynamics into discriminative learning for AI-generated video detection. Multiscale feature representation has been proven that it can to help improve the classification ability, for example, *Improved Pooling Attention* in MViTv2 (Li et al., 2022) shows its strong potential for building multiscale representations. We follow this basic setting and design the motion-aware spatiotemporal downsampling module to capture motion discrepancies between real and generated videos at different temporal scales. Specifically, we employ 3D convolutions to downsample both the temporal and spatial dimensions of features from the previous layer, as shown in Fig. 2(a).

Concretely, for the attention input at stage $i$, denoted as $\mathbf{x}_i \in \mathbb{R}^{L \times D}$, the sequence frame length is $L(L = T \times H \times W)$ with dimension $D$. We first project $\mathbf{x}_i$ linearly into queries $q_i \in \mathbb{R}^{L \times D}$, keys $k_i \in \mathbb{R}^{L \times D}$, and values $v_i \in \mathbb{R}^{L \times D}$. Before downsampling, we first reshape $q_i, k_i, v_i$ into spatial tensors of shape $T \times H \times W \times D$, and then apply 3D convolutions separately to obtain:

$$\hat{q}_{i+1} = \text{STDS}_{q_i}(q_i), \quad \hat{k}_{i+1} = \text{STDS}_{k_i}(k_i), \quad \hat{v}_{i+1} = \text{STDS}_{v_i}(v_i),$$

where $\text{STDS}_{q_i}, \text{STDS}_{k_i}, \text{STDS}_{v_i}$ denote spatiotemporal downsampling with 3D convolution operations. The convolution kernels and strides vary across downsampling stages, thereby producing outputs with different spatiotemporal scales. After downsampling, the results are reshaped back into sequences $q_{i+1}, k_{i+1}, v_{i+1}$ and processed with attention computation consistent with MViTv2, including spatiotemporal relative position embeddings and residual pooling connections. The relative position embedding $e^{(rel)}$ is computed by dot product between the current queries $q_{i+1}$ and dimension-adjusted embeddings from the original input. The procedure can be represented as:

$$\text{Attn}_{i+1} = \text{Softmax}\left(\frac{(q_{i+1}k_{i+1}^T + e^{(rel)})}{\sqrt{d}}\right)v_{i+1}, \tag{1}$$

$$e^{(rel)} = q_{i+1} \cdot r_t + q_{i+1} \cdot r_h + q_{i+1} \cdot r_w, \tag{2}$$

$$\mathbf{x}_{i+1} = \text{Attn}_{i+1} + q_{i+1}. \tag{3}$$

Through the spatiotemporal downsampling step, local spatiotemporal correlations from shallow layers are preserved through successive downsampling stages. At deeper layers with a larger spatiotemporal receptive field, these local cues are fused with global relations, which helps the model better learn the differences in motion patterns between real and generated videos. In addition, the spatiotemporal downsampling along $T$ not only reduces the computational cost and spatial complexity of attention but also improves the model's sensitivity to both global and local motion changes.

### 3.3 Cross-scale Semantic Contrastive Learning Module

For the AI-generated video detection task, another challenge is how to define the boundary of discrimination between the real and AI-generated videos. The goal of the supervised contrastive learning (Khosla et al., 2020) mechanism lies that features from the same class should be pulled closer, while those from different classes should be pushed apart in the embedding space. That is, the feature embedding similarity of two AI-generated videos should be higher than the feature embedding similarity of one AI-generated video and a real video. In addition, the number of samples in the sample set used in contrastive learning should be large enough to guide the feature learning. Therefore, we propose a novel Cross-scale Semantic Contrastive Learning (CSCL) module to enrich the diversity of the samples and guide the model learning more clear boundaries and discriminative features, as shown in Fig.2(b).

Specifically, the CSCL module is implemented on the last three stages of the pyramid. To enlarge the number of samples, we introduce the cross-scale features to enrich the sample set. The main reason lies in the fact that the feature representations of a video come from different scales should be semantically consistent. Based on this useful clue, we sample two blocks at each scale and randomly select one block. Then, we choose two different scales to feed into the supervised contrastive learning module. Specifically, the CSCL module is implemented on three different paired stages, *i.e.*, $\{\mathbf{x}_1, \mathbf{x}_2\}$, $\{\mathbf{x}_2, \mathbf{x}_3\}$, and $\{\mathbf{x}_1, \mathbf{x}_3\}$. Here, we use the paired stage $\{\mathbf{x}_1, \mathbf{x}_2\}$ as an example explanation and the two attention scale features are denoted as $\{x_1', x_2'\} \in \mathbb{R}^{L \times D}$. To ensure comparability across scales, the lower-dimensional features $x_2'$ are first projected into the same dimension as $x_1'$ via an attention mapping module. The sampled scale features serve as dimension-aligned contrastive groups. Although the features come from different stages, they originate from the same mini-batch of input data, which provides the basis for supervised contrastive learning. The supervised contrastive loss is defined as:

$$\mathcal{L}^{sup} = \sum_{i=1}^{N} \frac{-1}{|P_i|} \sum_{p \in P_i} \log \frac{\exp(\text{sim}(h_i, h_p)/\tau)}{\sum_{j=1}^{N} \exp(\text{sim}(h_i, h_j)/\tau)}, \tag{4}$$

where $h_i$ denotes the feature vector of sample $i$, $\text{sim}(h_i, h_p)$ is the cosine similarity between two vectors, $P_i = \{p \mid y_p = y_i, p \neq i\}$ is the set of indices of samples from the same class as $i$, $|P_i|$ is its cardinality, $N$ is the batch size, and $\tau$ is a temperature parameter. The final loss combines the cross-entropy loss $\mathcal{L}^{CE}$ and the supervised contrastive loss across three groups:

$$\mathcal{L} = \mathcal{L}^{CE} + \lambda\left(\mathcal{L}_1^{sup} + \mathcal{L}_2^{sup} + \mathcal{L}_3^{sup}\right). \tag{5}$$

Through the CSCL optimization, the model will be tend to learn class-discriminative features via the cross-scale consistency regularization.

# 4 AIDETECTION DATASET

**Organization:** To address the challenge of emerging new video generators, we propose a novel AIDetection dataset for the AI-generated video detection task. Table 1 presents the key differences between AIDetection and existing AI-generated video detection datasets. The main advantage of AIDetection lies in simultaneously providing a large number of generated videos from diverse sources and including the latest generation methods, which involve more sophisticated and heterogeneous mechanisms. This design makes it possible to evaluate the generalization performance of detectors trained on earlier generation models against the most recent ones. In addition, our collected real videos better match the content and style of videos in current online environments.

Table 1: Comparison of benchmark datasets for AI-generated video detection.

| Datasets | Scale | Latest model | Video sources (Gen./Real) |
|---|---|---|---|
| GVD (Bai et al., 2024) | 11k | Sora (2024.2) | 11/2 |
| GVF (Ma et al., 2024a) | 2.8k | Veo (2024.5) | 8/2 |
| GenVideo (Chen et al., 2024b) | 2271k | Sora (2024.2) | 24/3 |
| GenVidBench (Ni et al., 2025) | 143k | Mora (2024.3) | 8/2 |
| AIDetection | 39k | Hailuo (2025.6) | 24/3 |

The generated videos in AIDetection are partly sourced from the GVD (Bai et al., 2024) and Gen-Video (Chen et al., 2024b) datasets to construct a subset for our benchmark. Moreover, they are explicitly separated into I2V and T2V generation paradigms. Another portion of AIDetection comes from seven commercial generators (AI, 2024b; OpenAI, 2024; HiDream.ai, 2024; Jianying, 2024; AI, 2024a; WanTeam et al., 2025; KlingAI, 2024; PixVerse, 2024), where we collected publicly available demos from their official websites as well as user-uploaded community videos. Unlike the former category, these are mature commercial products that have undergone multiple iterations over several years and provide powerful customization capabilities. For example, Kling v1.6 not only supports traditional I2V and T2V modes, but also introduces functionalities such as first–last frame control, multi-image references, trajectory guidance, and multimodal editing on top of existing generations to replace or remove elements within a video. A portion of the real videos is sampled from the public action recognition datasets ActivityNet (Caba Heilbron et al., 2015) and Kinetics (Kay et al., 2017). Another portion is collected from publicly available popular videos on the Douyin short video platform. This design allows the AIDetection dataset to better simulate the distribution of real scenarios for video authentication, thereby making evaluation results more reliable.

**OOD Setting:** Given the specificity of AI-generated video detection, only detectors that generalize to arbitrary unseen generators or sources are meaningful. Following the domain generalization paradigm, we split videos by generator type or source: the source domain and target domain correspond to distinct sources used for training and testing, respectively. Domain shift arises from differences in generative texture characteristics, video quality, and semantic content. The exact training/test counts used in our evaluations are shown in the Appendix, which realistically simulates the need to discriminate a large number of unknown sources from a limited set of known ones.

# 5 EXPERIMENTS

**Experimental Setting:** In the preprocessing stage, we uniformly sample 16 frames from each raw video at equal temporal intervals. Each sampled frame is resized such that the shorter side is 256 pixels, followed by a random crop to a spatial resolution of $224 \times 224$. We then apply common data augmentation strategies, including *RandAugment* and *Random Erasing*. Our experiments are implemented with the `mmaction2` framework. The backbone adopts the "mvit-small" architecture, initialized with pretrained MViTv2 weights on Kinetics-400. Training is performed with a batch size of 16. For the supervised contrastive loss, the temperature parameter is set to $\tau = 0.1$ and the weighting coefficient $\lambda = 0.1$. We use the `AdamW` optimizer with an initial learning rate of $1.6 \times 10^{-4}$, momentum parameters $(\beta_1, \beta_2) = (0.9, 0.999)$, and weight decay of 0.05. All experiments

are conducted on $2 \times$ NVIDIA A800 80G GPUs.The learning rate is linearly warmed up from 1/10 of the base value during the first 20 epochs, and after that it is decayed with cosine annealing to 1/100 of the base value over the remaining epochs.

**Evaluations:** To comprehensively evaluate the generalization ability of the model when facing unseen generators and unknown real video sources, we conduct experiments on three datasets, *i.e.*, AIDetection, GVF (Ma et al., 2024a) and GenVideo (Chen et al., 2024b). Since the distributions are highly imbalanced across different sources, evaluating each generator individually would result in unstable distributions caused by varying sample sizes. Therefore, instead of following the original per-generator evaluation, we treat all test samples for evaluation. Although this change makes it difficult to compare our results directly with those of other methods, it provides a more realistic assessment of generalization to completely unknown distributions. Five metrics are used for evaluation, *i.e.*, ACC(%), Precision(%), Recall(%), F1(%), and Average Precision (AP). The accuracy calculation is based on a threshold value of 0.5.

Specifically, for the GVF dataset, we use 867 videos from each of ModelScopeT2V, Show1, Text2Video-zero, and ZeroScope, together with 867 real videos from MSR-VTT and MSVD as the training set. The test set includes videos from eight additional generators such as Pika and Sora, along with 97 real videos from MSR-VTT and MSVD. For the GenVideo dataset, due to its very large training set, we sample 18,405 generated videos from 10 generators and 19,806 real videos from 2 sources, which together account for about 1.7% of the entire dataset. The test set includes all 8,588 generated videos from 10 generators and 10,000 real videos from 1 source. For the one-to-many setting, we train on a single generator (OpenSora, Pika, or SEINE) with 2,572, 3,000, and 2,500 videos, respectively.

## 5.1 QUANTITATIVE EVALUATION

To comprehensively evaluate the generalization performance of our model when facing unseen generators and unknown real video sources, we conduct experiments on the AIDetection, GVF, and subsets of the GenVideo datasets. Our experimental settings on GenVideo and GVF differ from those in their original papers, making direct comparisons with the reported results infeasible. The detailed differences are provided in the Appendix. Table 2 presents the results when training with all available videos from the three datasets. MSTformer can achieve the best performance.

Table 2: Performance comparisons on three benchmark datasets.

| Dataset | Model | ACC | Precision | Recall | F1 | AP |
|---|---|---|---|---|---|---|
| AIDetection | UniFormerv2-B-(Li et al., 2023) | 86.13 | 80.92 | 94.56 | 87.21 | 95.30 |
| | MViTv2-S (Li et al., 2022) | 75.96 | 75.09 | 77.69 | 76.37 | 87.13 |
| | **MSTformer** | **91.31** | **93.10** | **89.23** | **91.12** | **97.08** |
| GenVideo | UniFormerv2-B-(Li et al., 2023) | 85.09 | 84.38 | 82.33 | 83.34 | 90.33 |
| | MViTv2-S (Li et al., 2022) | 78.24 | 94.43 | 55.24 | 69.70 | 89.74 |
| | **MSTformer** | **94.32** | **97.06** | **90.19** | **93.50** | **98.50** |
| GVF | **MSTformer** | **91.38** | **93.59** | **96.88** | **95.20** | **98.55** |

We further provide the detailed results of each generator to verify the effectiveness and balance of the proposed method. As shown in Table 3, eight advanced generators included in AIDetection are reported. To ensure the reliability of the results, the test samples here only consist of generated videos from the corresponding sources, excluding real videos. Therefore, we only report ACC(%) as the evaluation metric. These results provide strong evidence that MSTformer can efficiently generalize to multiple unseen video sources even with limited training data.

Table 3: Detailed ACC (%) comparison on 8 advanced generators from the AIDetection dataset.

| Model | Kling | PixVerse | Vidu | Jimeng | Hailuo | Wan | Sora | Hidream | Avg. |
|---|---|---|---|---|---|---|---|---|---|
| MViTv2-S | 94.33 | 88.03 | 75.45 | 45.00 | 65.57 | 87.41 | 71.74 | 81.79 | 81.79 |
| **MSTformer** | **97.38** | **95.87** | **89.70** | **80.33** | **86.53** | **96.53** | **68.60** | **86.26** | **86.26** |

**Analysis of one-to-many evaluation:** Moreover, we test the effectiveness of one-to-many evaluation setting according to the original advice of the GenVideo and GVF datasets. In Tables 4 and 5, we provide results trained on videos from a single generator to simulate more extreme generalization scenarios. Compared to training with multiple sources, the generalization performance drops significantly due to the limited diversity of motion patterns available for learning. Nevertheless, except in cases with very large inter-class differences, MSTformer still maintains competitive performance in most scenarios.

Table 4: One-to-many evaluation results of MSTformer on the GVF dataset.

| Training set | Text2Video-zero | Show1 | ModelScope | ZeroScope | Gen2 | Pika | Sora | Veo | Avg. |
|---|---|---|---|---|---|---|---|---|---|
| Text2Video-zero | 100.0 | 17.53 | 17.53 | 13.40 | 8.25 | 14.43 | 60.42 | 21.43 | 21.43 |
| Show1 | 4.12 | 96.91 | 78.35 | 73.20 | 96.91 | 89.69 | 22.92 | 85.71 | 85.71 |
| ModelScope | 2.06 | 84.54 | 95.88 | 94.85 | 97.94 | 89.69 | 35.42 | 64.29 | 64.29 |
| ZeroScope | 69.07 | 70.10 | 87.63 | 100.0 | 94.85 | 74.23 | 56.25 | 92.86 | 92.86 |
| Full training set | 100.0 | 100.0 | 96.91 | 98.97 | 98.97 | 94.85 | 75.00 | 92.86 | 92.86 |

Table 5: Comparison of different methods on One-to-many testing results in the GenVideo dataset.

| Training set | Model | Sora | MorphStudio | Gen2 | HotShot | Lavie | Show1 | MoonValley | Crafter | ModelScope | WildScrape | Avg. |
|---|---|---|---|---|---|---|---|---|---|---|---|---|
| OpenSora | UniFormerV2-B | 21.43 | 67.14 | 73.77 | 79.14 | 77.57 | 71.86 | 85.46 | 74.32 | 47.71 | 49.84 | 64.82 |
| | **MSTformer** | 19.64 | 71.57 | 95.29 | 94.00 | 83.93 | 86.43 | 93.93 | 88.63 | 65.86 | 75.24 | 77.45 |
| Pika | UniFormerV2-B | 39.29 | 61.86 | 80.12 | 44.29 | 57.07 | 53.29 | 86.90 | 75.54 | 43.29 | 52.56 | 59.42 |
| | **MSTformer** | 14.29 | 69.86 | 97.25 | 92.71 | 76.64 | 85.43 | 91.37 | 86.84 | 53.29 | 65.37 | 73.31 |
| SEINE | UniFormerV2-B | 41.07 | 84.71 | 89.57 | 84.43 | 74.29 | 65.57 | 91.21 | 91.13 | 65.00 | 56.87 | 74.39 |
| | **MSTformer** | 28.57 | 70.57 | 93.55 | 91.57 | 83.71 | 81.00 | 76.29 | 89.06 | 76.29 | 62.78 | 75.34 |
| Full Set | UniFormerV2-B | 30.36 | 82.43 | 90.80 | 82.71 | 78.43 | 78.00 | 95.37 | 92.42 | 73.71 | 57.99 | 76.22 |
| | **MSTformer** | 26.79 | 88.86 | 98.62 | 83.71 | 87.14 | 92.29 | 98.24 | 97.57 | 82.14 | 74.92 | 83.03 |

We can also find that MSTformer is consistently robust across heterogeneous OOD tasks: in AIDetection, the test set is far more diverse than the training set; in GenVideo one-to-many evaluation, the training set covers only one generator and is smaller than the test set. In both cases, MSTformer maintains strong performance, indicating effective generalization from limited sources to unseen, higher-diversity videos. Performance on GVF is relatively lower, likely due to small and highly imbalanced test subsets, which may introduce estimation bias.

## 5.2 ABLATION STUDY

**Ablation Study on MSTD and CSCL.** To directly verify the effect of introducing motion-aware spatiotemporal downsampling (MSTD) and cross-scale semantic contrastive learning (CSCL) on model generalization, we design a challenging out-of-distribution (OOD) task and conduct an ablation study on the baseline method MViTv2. The training set only contains three types of generated videos (SEINE, Lavie, and OpenSora) and two subsets of real-video datasets (ActivityNet and Kinetics), consisting of 6,800 video samples in total. The test set includes 4,222 videos sampled from all other generators and real videos from Douyin. To ensure fairness, we keep almost all training parameters and strategies identical to the baseline video classification tasks, and train the model from scratch. Data augmentation and cropping strategies are also kept consistent. For frame sampling, we follow the default setting of MViTv2, which uniformly samples 16 frames with a fixed interval of 4 to form continuous video clips.

The ablation results are shown in Table 6. After adding the motion-aware spatiotemporal downsampling (MSTD) mechanism, the recall score shows a significant improvement on the test set, which increases from 46.56% to 66.81%. The main improvement demonstrates that video temporal cues can effectively reduce misjudgment where the model tends to predict all videos as real under complex spatial semantics, thereby enhancing generalization. It also indicates that motion semantics along the temporal dimension play a crucial role in distinguishing generated videos from real ones. If we further employ the cross-scale semantic contrastive learning (CSCL) module, the performance will be enhanced. Since the CSCL module introduces the motion-aware features across different scales, it can not only alleviate the confusion over different video semantic representations but also prevent the model's performance on real-video discrimination from degrading. We also analyze the effect of the temperature parameter $\tau$ in the contrastive loss of CSCL. If we set $\tau$ as a very small

value (e.g., 0.07), the model training causes it to diverge, with the loss quickly becoming NaN. This is because a small $\tau$ excessively amplifies the logits in the softmax function, leading to numerical overflow in the exponential operation and extremely large gradients, which destabilize optimization. In contrast, moderate values of $\tau$ (e.g., 0.1–0.2) achieve a balance between discrimination and stability, yielding the best performance in our experiments.

Table 6: Ablation results on baseline MViTv2. MST-DS: Motion-aware Spatiotemporal Downsample, MST-CL: Cross-scale Semantic Contrastive Learning.

| Method | MSTD | CSCL | ACC | Precision | Recall | F1 |
|--------|------|------|-----|-----------|--------|-----|
| MViTv2 | – | – | 69.56 | 86.23 | 46.56 | 60.47 |
| MViTv2 | ✓ | – | 75.27 | 80.43 | 66.81 | 72.99 |
| MViTv2 | ✓ | $(\lambda = 0.2, \ \tau = 0.1)$ | 78.33 | 89.60 | 64.10 | 74.73 |
| MViTv2 | ✓ | $(\lambda = 0.1, \ \tau = 0.2)$ | 77.77 | 90.29 | 62.24 | 73.69 |
| MViTv2 | ✓ | $(\lambda = 0.1, \ \tau = 0.1)$ | 77.83 | 85.17 | 67.39 | 75.24 |

**The effect of Sampling Frames:** Because video lengths are various, the number of frames needed to sample from the original video naturally affects the performance of classifiers. Therefore, we conduct an ablation study on the sampling length of MSTformer when training and testing on the AIDetection dataset under OOD settings, as shown in Fig. 3. The results show that increasing the number of sampled frames yields a clear improvement in terms of multiple metric scores. This indicates that the model benefits from capturing richer and more robust motion patterns from longer temporal sequences, which demonstrates both its ability to fully exploit temporal and motion cues as well as its potential when more computational resources are available.

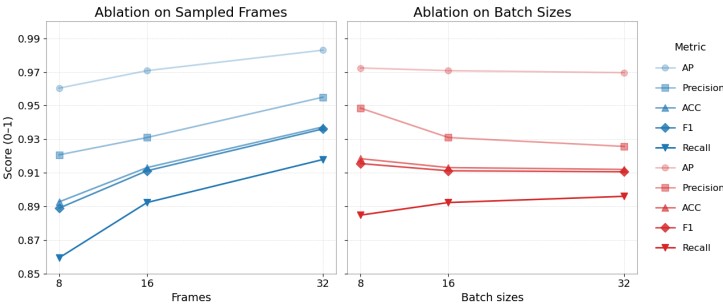

Figure 3: Impact of frame sampling and batch sizes on AIDetection OOD results

**The effect of Batch Sizes:** Supervised contrastive learning is inevitably influenced by the number of positive and negative samples within each minibatch. As shown in Fig. 3, MSTformer does not show significant performance differences under varying batch sizes. However, a large batch size will bring more computational burden, hence we choose 16 as the batchsize for further training. It is worth mentioning that our proposed cross-scale contrastive group can provide sufficient positive and negative samples even if the batch size is small, this design help the contrastive learning can converge to a steady state.

## 6 CONCLUSION

In this paper, we address the poor generalization of existing detectors to high-quality AI-generated videos by identifying intrinsic motion-pattern discrepancies in the generative process. Building on this insight, we construct AIDetection, a dataset tailored to realistic deployment scenarios. We propose a novel lightweight detector MSTformer, that (i) employs motion-aware spatiotemporal downsampling to capture local motion differences between the AI-generated and real videos, and (ii) introduces cross-scale semantic contrastive learning to enforce cross-scale feature consistency and mitigate motion-cue confusion. Experiments show that MSTformer is robust under OOD settings across AIDetection, GVF, and GenVideo. We hope this work shifts the community's focus from brittle spatial artifacts toward motion-centric principles for AI-generated video detection.

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

## A   APPENDIX

### A.1   AI-GENERATED VIDEO DETECTION BENCHMARKS

To address this emerging challenge, several datasets and evaluation benchmarks dedicated to AI-generated video detection have been proposed. Bai et al. (2024) introduced the Generated Video Dataset (GVD), which contains 11,618 generated video samples collected from 11 mainstream models. The dataset is divided into two categories, text-to-video (T2V) and image-to-video (I2V). It covers diverse scenarios such as human activities, natural environments, and object motion. Ma et al. (2024a) constructed GVF, the first public benchmark dataset for generated video detection. GVF consists of 964 real videos sampled from MSVD (Chen & Dolan, 2011) and MSR-VTT (Xu et al., 2016), and their corresponding prompts and generated counterpart videos using four different models: Text2Video-zero, ModelScopeT2V, ZeroScope, and SHOW-1. To mitigate the lack of large-scale, high-quality datasets for this field, Chen et al. (2024b) released GenVideo, the first million-scale dataset for AI-generated video detection. It categorizes data by generator type, with training videos generated by 10 models along with real videos from Kinetics-400 (Kay et al., 2017) and Youku-mPLIG (Xu et al., 2023), while the test set includes videos from another 10 generators and real videos from MSR-VTT (Xu et al., 2017). Based on this design, the authors proposed cross-generator classification tasks and degraded video classification tasks to evaluate generalization and robustness. More recently, Ni et al. (2025) introduced GenVidBench, a benchmark of over 100,000 videos. Similar to GenVideo, it partitions training and test sets by generator categories. In addition, GenVidBench separates training and testing splits according to prompts or input images, and introduces cross-source real/fake classification tasks to evaluate the ability of detectors to generalize without relying on generator-specific cues.

### A.2   AIDETECTION DATASET

As next-generation generators rapidly improve, we construct AIDetection to train detectors and evaluate OOD generalization to unseen sources. All video sources and related information are summarized in Table 7.

The generated videos in AIDetection are partly sourced from the GVD (Bai et al., 2024) and Gen-Video (Chen et al., 2024b) datasets, where we sampled specific categories (Ma et al., 2024b; AI, 2023a; Labs, 2023; AI, 2023b; Studio, 2023; Kondratyuk et al., 2024; Girdhar et al., 2023; Chen et al., 2024a; Wang et al., 2025; Zheng et al., 2024; Xing et al., 2024; Blattmann et al., 2023; Chen et al., 2023) to construct a subset for our benchmark. These videos employ generation technologies popular between 2023 and 2024, including both autoregressive and diffusion-based models. They cover a wide range of semantic contents and styles, such as realistic humans and animals, static scenes or objects, cartoon characters, and even surreal artistic imagery. Moreover, they are explicitly separated into I2V and T2V generation paradigms.

Another portion of AIDetection comes from seven commercial generators (AI, 2024b; OpenAI, 2024; HiDream.ai, 2024; Jianying, 2024; AI, 2024a; WanTeam et al., 2025; KlingAI, 2024; Pix-Verse, 2024), where we collected publicly available demos from their official websites as well as user-uploaded community videos. Unlike the former category, these are mature commercial products that have undergone multiple iterations over several years and provide powerful customization capabilities. For example, Kling v1.6 not only supports traditional I2V and T2V modes, but also introduces functionalities such as first–last frame control, multi-image references, trajectory guidance, and multimodal editing on top of existing generations to replace or remove elements within a video. Such advanced features result in greater diversity of generation strategies, making it difficult to trace back the exact generation pipeline (labeled as "Unknown" in the *Task* column of Table 7). Including this type of video not only enriches the diversity of the training data but also ensures that evaluation scenarios are better aligned with real-world social media applications. The distribution of the main semantic content in these generated videos is shown in the fig 4.

A portion of the real videos is sampled from the public action recognition datasets ActivityNet (Caba Heilbron et al., 2015) and Kinetics (Kay et al., 2017), which cover hundreds of complex human activity categories and represent the majority of real human-centered actions. Another portion is collected from publicly available popular videos on the Douyin short video platform. We carefully removed samples that may contain AI-generated manipulations or that were stitched from static im-

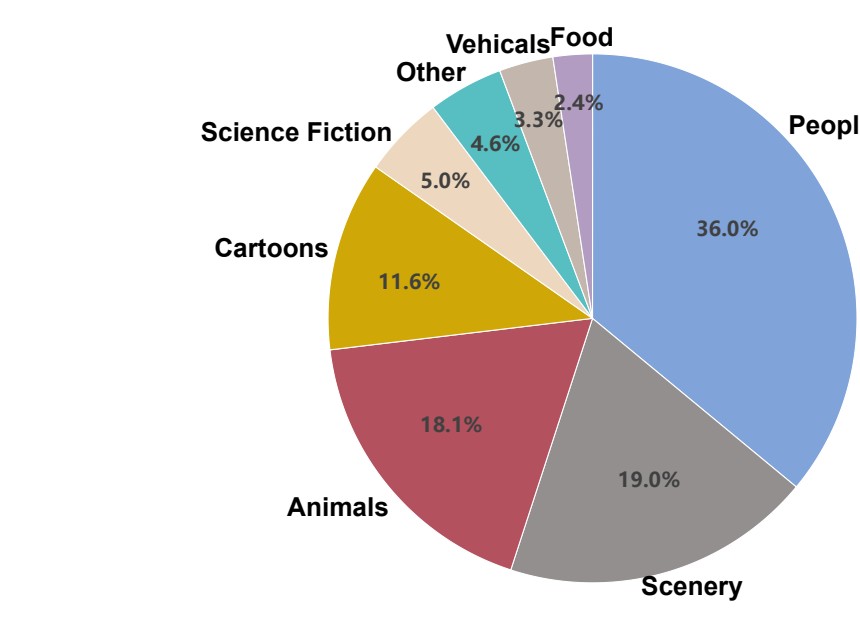

Figure 4: Distribution of the main semantic content in generated videos.

ages. The Douyin videos include diverse types of real-world content such as news reports, movie clips, and documentaries, and their subjects extend beyond human activities to animals, objects, and natural scenes. This design allows the AIDetection dataset to better simulate the distribution of real scenarios for video authentication, thereby making evaluation results more reliable.

**OOD Setting:** Classical machine learning typically assumes that the source and target domains are i.i.d. In practice, however, domain shift leads to OOD scenarios. Domain generalization (DG) aims to learn solely from source-domain data and generalize to unseen target domains (Yao et al., 2021; Lin et al., 2023). Given the specificity of AI-generated video detection, only detectors that generalize to arbitrary unseen generators or sources are meaningful.

Following the DG paradigm, we split videos by generator type or source: the source domain and target domain correspond to distinct sources used for training and testing, respectively. Domain shift arises from differences in generative texture characteristics, video quality, and semantic content. The exact training/test counts used in our evaluations (Section 5.1 are reported in Table 7, which realistically simulates the need to discriminate a large number of unknown sources from a limited set of known ones. Thanks to its diverse generator coverage, AIDetection supports flexible sampling via label files to construct both OOD and in-domain tasks, enabling training and evaluation from multiple perspectives—one of the dataset's key advantages.

### A.2.1 VISUALIZATION OF AIDETECTION DATASET

Fig. 5–12 shows visualizations of 8 categories of advanced generated videos included in the AIDetection dataset.

### A.3 ADDITIONAL INFERENCE RESULTS

To further validate the generalization capability of **MSTformer**, we present inference results on a subset of generated and real videos in Fig 13. Some of these videos are sampled from the **AIDetection** dataset, while others are obtained from publicly available videos on different media platforms. In the figures, we annotate the authenticity of each video, its source, and the probabilities of being classified as generated or real videos, denoted as $[p_{generated}, p_{real}]$.

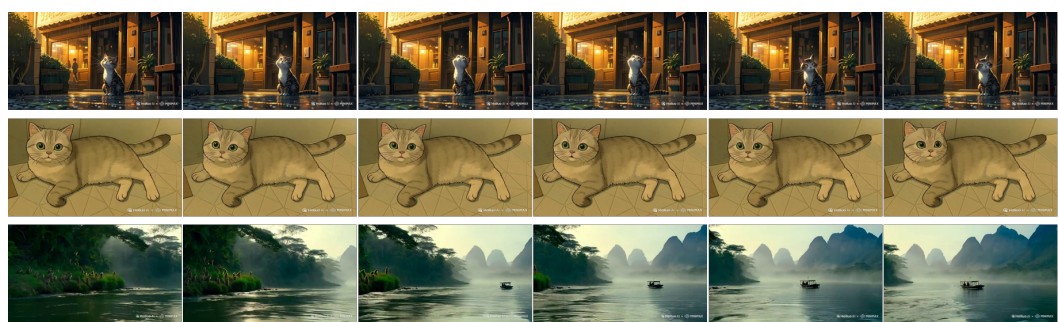

Figure 5: Hailuo generated samples visualization.

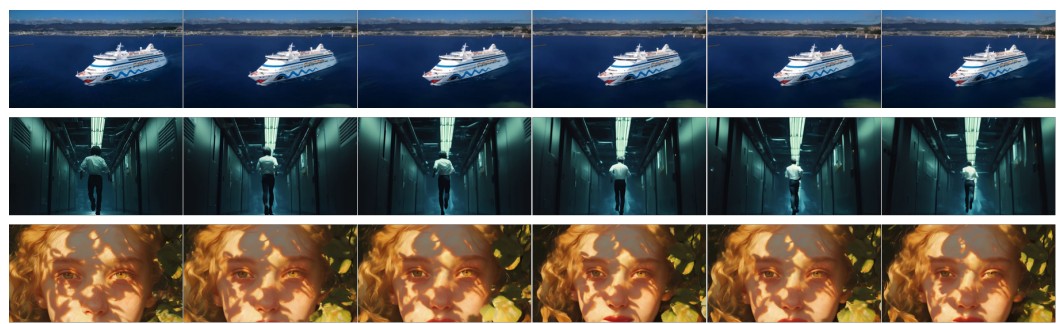

Figure 6: Jimeng generated samples visualization.

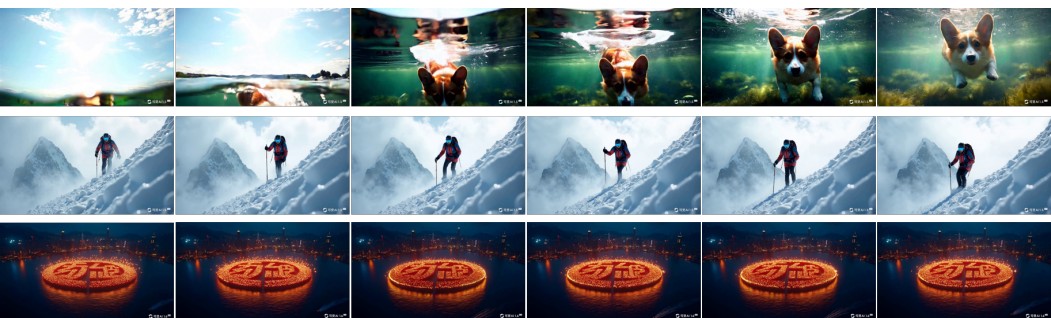

Figure 7: Kling generated samples visualization.

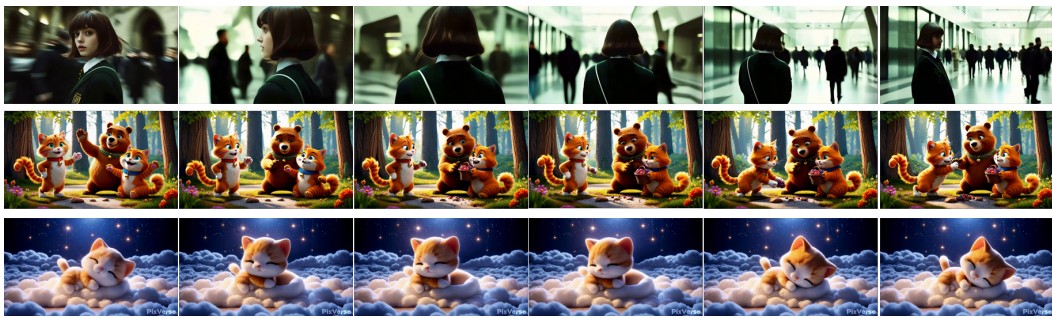

Figure 8: PixVerse generated samples visualization.

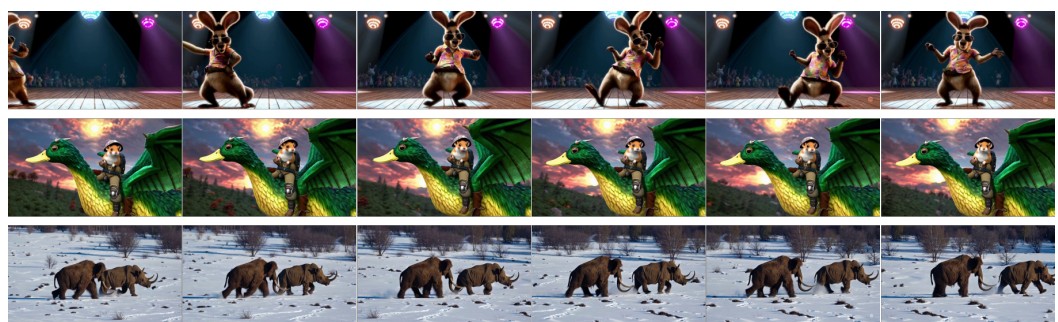

Figure 9: Sora generated samples visualization.s

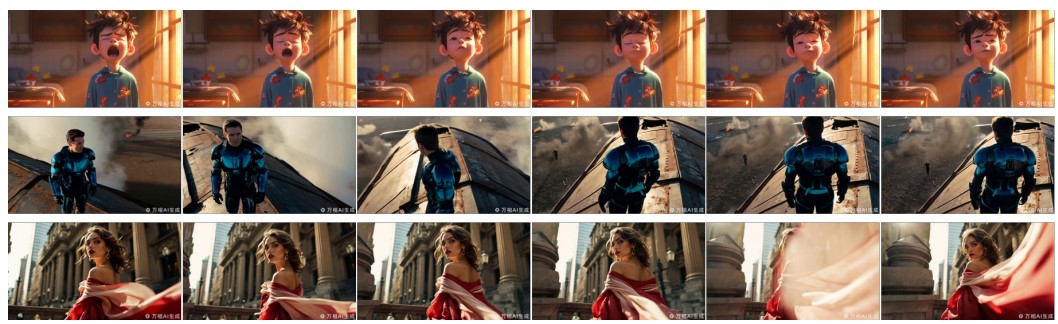

Figure 10: Wan generated samples visualization.

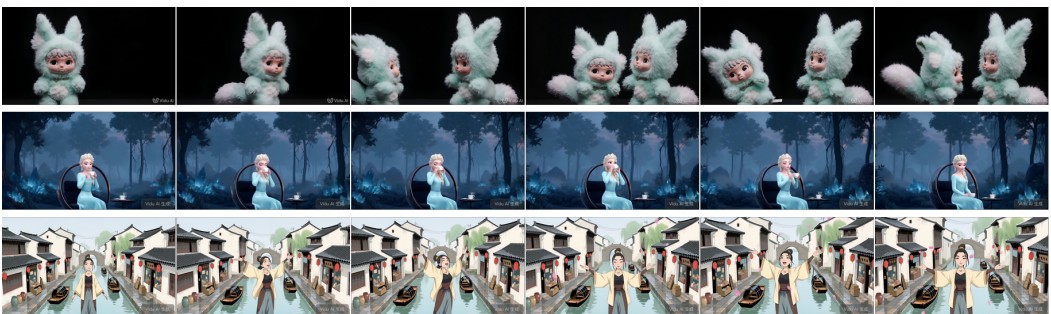

Figure 11: Vide generated samples visualization.

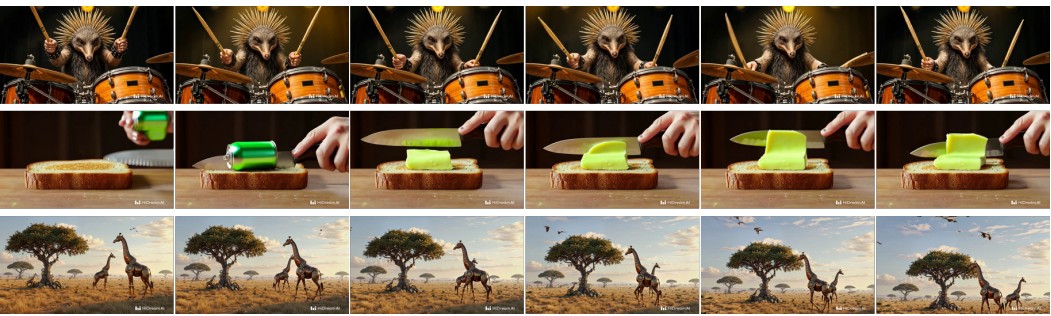

Figure 12: Hidream generated samples visualization.

Table 7: Overview of all video sources in the AIDetection dataset

| Type | Task | Source | Resolution | Fps | Length | Quantity | Training set | Testing set |
|------|------|--------|-----------|-----|--------|----------|--------------|-------------|
| Generated | T2V | Latte | 256×256 | 8 | 2s | 1000 | – | 300 |
| | | Moonvalley | 1184×672 | 16 | 5s | 1000 | – | 300 |
| | | Pika | Variable | 24 | 3s | 1000 | – | 300 |
| | | NeverEnds | Variable | 8–10 | 2–4s | 1000 | – | 300 |
| | | MorphStudio | 1024×576 | 8 | 2s | 700 | – | 300 |
| | | VideoPoet | Variable | 8 | 2–16s | 120 | – | 100 |
| | | Emu | 512×512 | 16 | 4s | 900 | – | 300 |
| | | VideoCrafter | 1024×576 | 8 | 2s | 1500 | – | 300 |
| | | Lavie | 512×320 | 8–24 | 2s | 1400 | 1400 | – |
| | | OpenSora | 256×256 | 8 | 2s | 1000 | 1000 | – |
| | I2V | DynamicCrafter | 1024×576 | 8 | 2s | 1000 | – | 300 |
| | | Moonvalley | Variable | 16–50 | 1–3s | 1000 | – | 300 |
| | | Pika | Variable | 8–24 | 3s | 1000 | – | 300 |
| | | SVD | 1024×576 | 7 | 3s | 1000 | – | 300 |
| | | SEINE | 1024×576 | 8 | 2s | 1000 | 1000 | – |
| | | NeverEnds | Variable | 10 | 3s | 1000 | 1000 | – |
| | Unknown | Vidu | Variable | 16–120 | 2–74s | 505 | – | 300 |
| | | Sora | Variable | 30 | 3–60s | 605 | – | 300 |
| | | Hidream | Variable | 16–32 | 3–12s | 313 | – | 300 |
| | | Jimeng | Variable | 8–60 | 2–90s | 300 | – | 300 |
| | | Hailuo | Variable | 24–30 | 5–106s | 334 | – | 300 |
| | | Wan | Variable | 25–30 | 4–8s | 548 | – | 300 |
| | | Kling | Variable | 24–30 | 5–10s | 917 | 800 | – |
| | | PixVerse | Variable | 15–60 | 3–28s | 702 | 700 | – |
| Real | – | ActivityNet | Variable | 6–30 | 3–15s | 2299 | 2000 | – |
| | | Kinetics | Variable | 30 | 1–10s | 1000 | 4000 | – |
| | | Douyin | Variable | 10–60 | 4–240s | 6999 | – | 5200 |

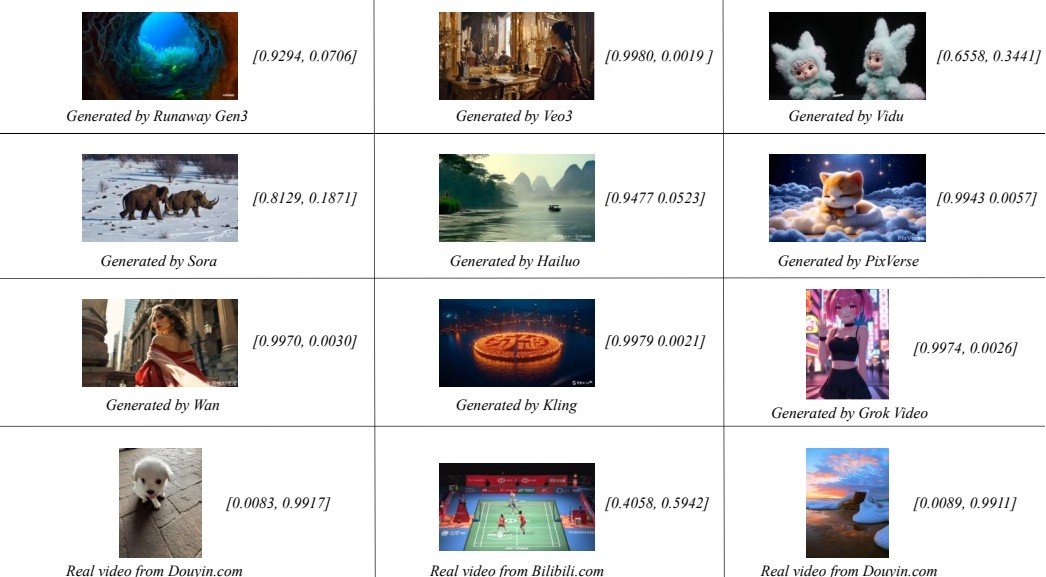

Figure 13: Inference results of MSTformer on diverse generated and real videos.

The stride is defined as the total number of frames divided by 16. This ensures that the input frames cover the entire temporal span of the video. Finally, we normalize the pixel values using channel-wise mean $[114.75, 114.75, 114.75]$ and standard deviation $[57.375, 57.375, 57.375]$.

## B    LLM Usage Statement

In preparing this manuscript, we employed a large language model (LLM) solely for language polishing and minor grammatical improvements. The LLM was not used for research ideation, data analysis, experimental design, or content generation. All technical content, methodologies, and conclusions are the sole responsibility of the authors.

