# Supplementary Material: Temporal Optical Flow Analysis

## Overview

In this supplementary material, we visualize the temporal evolution of optical flow magnitude across different video sources. The comparison is divided into two parts: real-world datasets and generative video models. In all plots, the **blue line** represents the average optical flow speed, and the **pink shaded area** indicates the interquartile range (IQR), reflecting motion variance.

## 1 Real-World Datasets Baseline

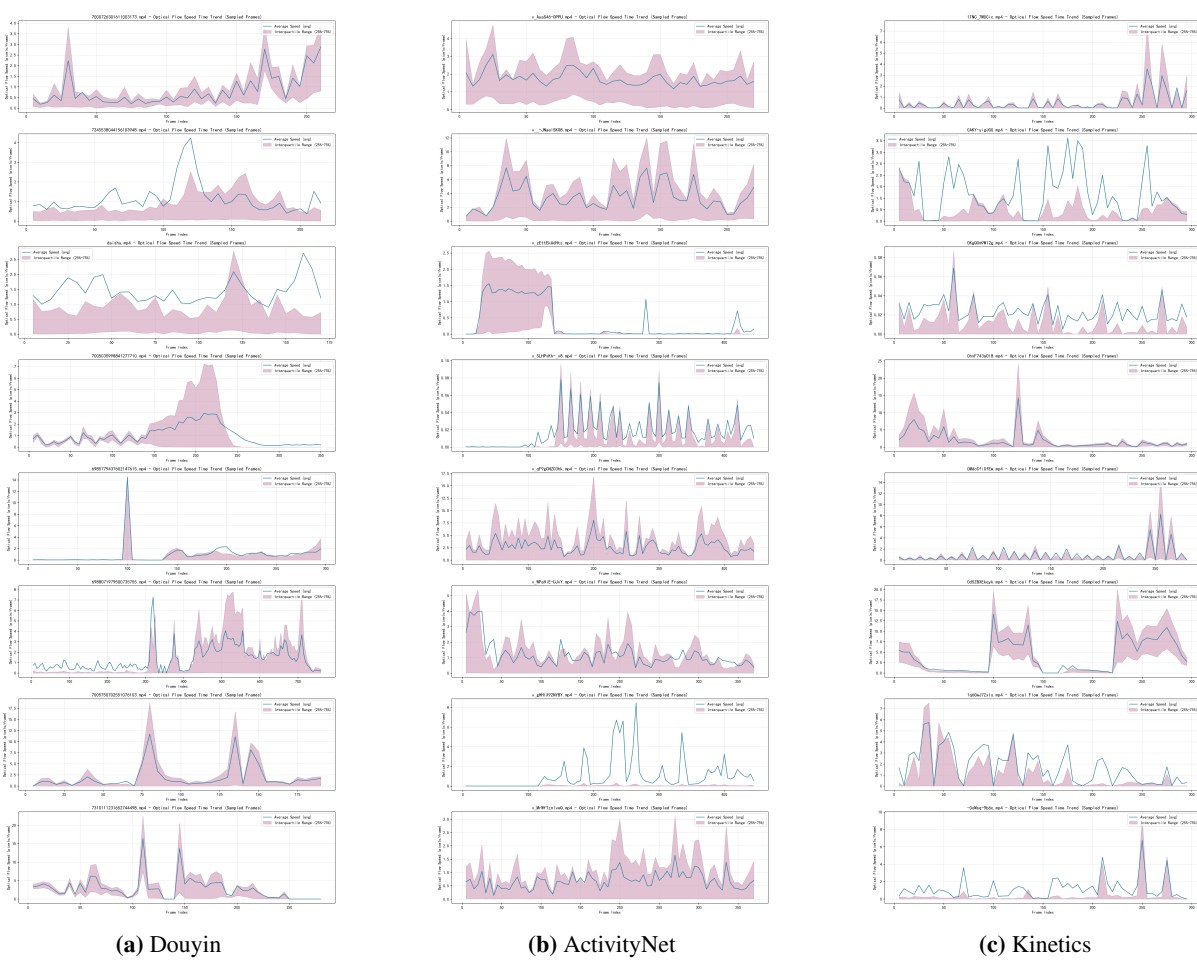

**(a)** Douyin  **(b)** ActivityNet  **(c)** Kinetics

**Figure 1: Real-World Motion Patterns.** These plots illustrate the ground truth optical flow trends used as a reference for natural motion.

**Analysis.** We first establish a baseline using three diverse real-world datasets: **Douyin**, **ActivityNet**, and **Kinetics**. As shown in Figure 1, real-world videos exhibit complex and highly variable motion

patterns. Specifically, action-oriented datasets (ActivityNet, Kinetics) show distinct peaks corresponding to human movements, while Douyin displays a wide dynamic range typical of in-the-wild recordings.

## 2    Analysis on Generative Video Models

**Analysis.**    In this section, we compare the optical flow dynamics of six generative models, categorized into leading SOTA models (**Sora, Hailuo, Wan**) and other baselines (**Hidream, Jimeng, NeverEnds**).

As illustrated in Figure 2, we observe distinct motion characteristics:

- **Stability vs. Dynamics (Top Row): Sora** and **Hailuo** demonstrate exceptional temporal stability, characterized by smooth optical flow curves and consistent variance (pink area). In contrast, **Wan** exhibits a larger dynamic range with higher peaks, reflecting motion patterns that are closer to the high-variance nature of real-world videos.

- **Motion Artifacts and Diversity (Bottom Row):** The baseline models show varied behaviors. **NeverEnds** displays flatter or periodic trends, which aligns with the characteristics of infinite-loop generation. **Hidream** and **Jimeng** exhibit irregular fluctuation patterns, highlighting the differences in how their respective diffusion backbones enforce temporal coherence.

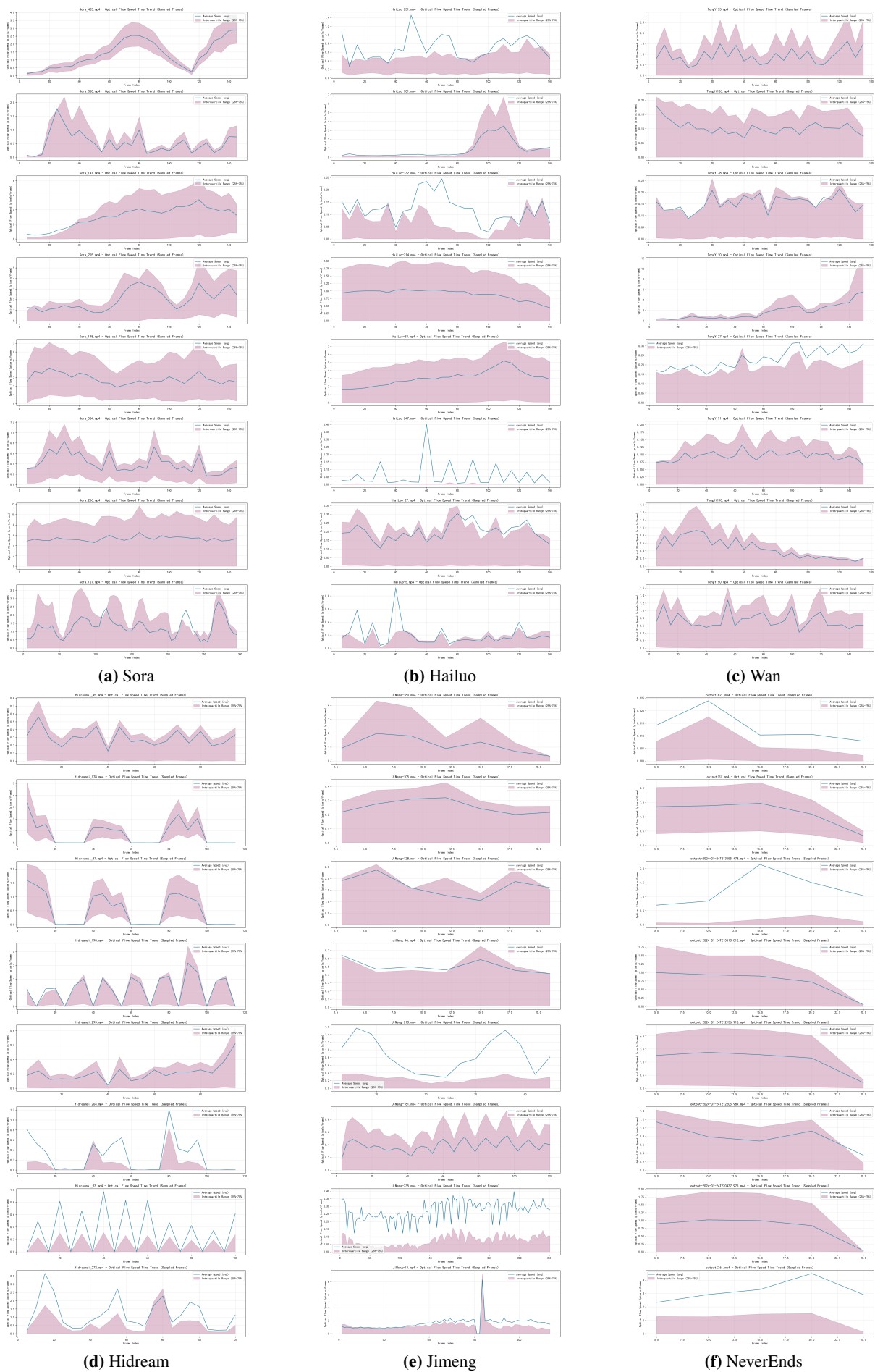

**Figure 2: Temporal Dynamics of Generative Models.** The top row displays leading SOTA models characterized by high stability or realistic dynamic ranges. The bottom row illustrates other baselines with diverse motion signatures and variance patterns.