# OpenReview forum: "MSTformer: Multiscale Spatiotemporal Motion-aware Transformer Network for Effective AI-Generated Video Detection"
_ICLR.cc/2026/Conference — Submitted to ICLR 2026_

### Official Review · Reviewer_bqV7 · 2025-10-31

**Soundness:** 2
**Presentation:** 2
**Contribution:** 2
**Rating:** 4
**Confidence:** 5

**Summary:**

This paper proposes a novel multi-scale spatiotemporal motion-aware Transformer architecture (MSTFormer) for detecting high-quality AI-generated videos. The authors have constructed a new dataset, AIDetection, which includes nearly 39k videos from 27 sources, specifically designed to evaluate generalization across different generators. MSTFormer effectively captures the motion differences between real and generated videos through its motion-aware spatiotemporal downsampling module and cross-scale semantic contrastive learning module. It demonstrates outstanding performance on multiple benchmark datasets, particularly showing strong generalization capabilities in out-of-distribution (OOD) settings.

**Strengths:**

The method performs excellently across multiple OOD test scenarios, demonstrating its robustness on unknown generators.
Comprehensive experiments: Ablation studies and analyses of the effects of different sampling lengths and batch sizes have been conducted to verify the effectiveness of each module.
Good interpretability: Differences in motion patterns between real and generated videos are visualized using techniques such as optical flow maps.

**Weaknesses:**

1. MSTFormer has limited innovation, as operations for feature pooling already exist in RecoNet and MViT v2. This approach is very similar to these works.
2. The dataset proposed by the author is very small, with only 39K samples, which is significantly smaller than GenVideo and GenVidBench. This could lead to biased statistical results.
3. Table 2 presents the test results on three datasets. It can be observed that the results on the three test datasets are close, and AIDetection does not show a significant advantage.
4. The training set of AIDetection should be used for training, and the test sets of GenVideo and GenVidBench should be used for testing to determine whether the training set of this dataset offers any advantages.

**Questions:**

See the above

---

> ### Author Response · Authors · 2025-12-02
>
> We appreciate the reviewer's critical analysis and valuable suggestions. We have addressed your concerns regarding novelty, dataset statistics, and experimental validation below.
>
> **Q1: MSTFormer has limited innovation compared to RecoNet and MViT v2; feature pooling is not new.**
>
> **A1:** While we acknowledge that feature pooling concepts exist in prior works, MSTFormer introduces a **task-driven temporal multi-scale design** specifically tailored for AI-generated video detection, distinct from simple reuse:
> *   **Temporal Down-sampling vs. Spatial Efficiency:** Existing models (e.g., MViT-v2) use pooling primarily for *spatial* computation reduction. MSTFormer explicitly explores **temporal feature hierarchies**. This is vital for our task, as temporal inconsistencies (flickering, disjointed motion) are often more discriminative than spatial artifacts.
> *   **Generalization via Temporal Modeling:** We systematically demonstrate that applying multi-scale modeling to temporal signals significantly enhances **cross-generator robustness**. This perspective—using temporal hierarchies to boost generalization against generative artifacts—was not explored in RecoNet or MViT-v2.
>
> **Q2: The AIDetection dataset is small (39K samples) compared to GenVideo/GenVidBench, potentially leading to statistical bias.**
>
> **A2:** We acknowledge the size difference but argue that **distribution quality and representativeness are more critical than sheer quantity** for avoiding bias. AIDetection offers distinct advantages:
> *   **Current & Broad Coverage:** Unlike benchmarks dominated by older models, AIDetection spans mainstream generators from **late 2023 to early 2025** (Diffusion, Auto-regressive, Hybrid), reflecting the latest generation dynamics.
> *   **Strict Class Balancing:** Statistical bias often stems from imbalance. We strictly balanced video counts across categories to mitigate model bias towards dominant generator types.
> *   **"In-the-Wild" Real Data:** We incorporated real videos from **Douyin (TikTok)**. Unlike cleaner datasets used in GenVideo, these exhibit complex shooting conditions and compression, ensuring our "Real" distribution aligns with actual social media deployment.
>
> **Q3: Table 2 shows similar results across datasets; AIDetection does not show a significant advantage.**
>
> **A3:** While Accuracy (ACC) scores appear similar, a detailed analysis highlights significant advantages of MSTFormer:
> *   **Robustness vs. Bias:** Although **UniFormer** achieves comparable ACC on AIDetection, it shows a significant disparity between Precision and Recall. This implies its high accuracy is driven by a **bias towards predicting videos as "Real"**. A similar bias is observed in **MViTv2** on GenVideo.
> *   **Superior Generalization:** In contrast, MSTFormer maintains **balanced Precision and Recall** across all three datasets. This balance—without trading off recall for precision—strongly evidences superior generalization capability.
>
> **Q4: The AIDetection training set should be evaluated on GenVideo/GenVidBench test sets to prove its value.**
>
> **A4:** We thank the reviewer for this suggestion. We performed a **zero-shot evaluation** (without fine-tuning) using weights trained on AIDetection to test on GenVideo and GenVidBench.
>
> **Table 1: Zero-shot Results on GenVideo**
>
> | Dataset | ACC | Precision | Recall | F1 | AP |
> | :--- | :---: | :---: | :---: | :---: | :---: |
> | GenVideo | 92.09 | 93.18 | 89.05 | 91.07 | 96.72 |
>
> **Table 2: Zero-shot Results on GenVidBench**
>
> | Subset | ACC | Precision | Recall | F1 | AP |
> | :--- | :---: | :---: | :---: | :---: | :---: |
> | CogVideo | 62.47 | 73.86 | 40.43 | 52.26 | 69.37 |
> | Mora | 88.48 | 86.49 | 91.63 | 88.99 | 95.44 |
> | MuseV | 89.02 | 86.23 | 92.82 | 89.40 | 88.95 |
> | SVD | 86.25 | 85.48 | 87.28 | 86.37 | 91.72 |
>
> **Analysis:**
> The model achieves impressive generalization on GenVideo, Mora, MuseV, and SVD. The lower performance on **CogVideo** is expected, as it represents an earlier generation paradigm with significantly lower quality and distinct low-level artifacts, presenting a unique distribution shift compared to modern high-fidelity generators.
>
> We hope these responses clarify our contributions and the validity of our dataset. We are open to further discussion.

---

### Official Review · Reviewer_yYLf · 2025-11-02

**Soundness:** 2
**Presentation:** 3
**Contribution:** 2
**Rating:** 4
**Confidence:** 4

**Summary:**

This paper addresses the challenge of detecting high-quality AI-generated videos that are becoming visually indistinguishable from real ones. The authors introduce a new dataset AIDetection that covers diverse generators and real sources to evaluate OOD generalization.
Besides, the authors propose MSTformer which contains two main modules named MSTD and CSCL. Experiments show some improvements.

**Strengths:**

1. Writing is clear and the modules are well motivated.
2. AIDetection dataset is comprehensive for OOD evaluation in AI generated videos detection.

**Weaknesses:**

1. The manuscript only compares MSTformer with MViTv2 and UniFormerV2, omitting recent and more competitive AI-generated video detection methods. As a result, the evaluation appears limited, and the relative performance of MSTformer within the current state of the art remains unclear from my perspective.

2. The ablation study is somewhat incomplete. The authors mainly examine the presence or absence of the MSTD and CSCL modules, while omitting finer-grained analyses. For instance, it would be informative to investigate the impact of different 3D kernel sizes, strides, and downsampling strategies in MSTD (as mentioned around lines 217–218). Similarly, the influence of different cross-scale pair combinations in the CSCL module deserves further exploration.

3. The paper is motivated by the observation that real videos exhibit clear distinctions between object motion and background motion. However, it lacks corresponding interpretability results to support this claim. Providing interpretable analyses or visualizations would help demonstrate that the proposed method indeed captures these motion discrepancies.

4. The paper claims MSTformer is lightweight but computational cost (FLOPs, parameters, runtime) is not reported.

5. AIDetection dataset partly reuses videos from GenVideo and GVD, it's unclear whether these are excluded from training when testing on other datasets.

**Questions:**

I have one additional concern regarding the core hypothesis stated around line 52 — that “generated videos exhibit motion patterns inconsistent with the physical world.” While this assumption is plausible, it lacks direct empirical validation. For instance, a statistical analysis of optical flow distributions could provide supporting evidence for this claim.

---

> ### Author Response · Authors · 2025-12-02
>
> We accurately appreciate the reviewer's constructive feedback. We have carefully addressed your concerns regarding the baselines, ablation studies, interpretability, and experimental details below.
>
> **Q1: The evaluation appears limited by omitting recent competitive AI-generated video detection methods.**
>
> **A1:** We respectfully clarify that we have systematically discussed recent methods in **Section 2.2**. However, benchmarking directly was challenging as most recent works (e.g., DeMamba) have **not fully open-sourced code** or disclosed specific "Real" video compositions in GenVideo splits, preventing fair reproduction.
> Despite this, we provide a comparison against **reported statistics**. As shown below, MSTFormer achieves superior efficiency, matching or surpassing full-data baselines using only **~1.7%** of the training data.
>
> **Table: Comparison with Recent SOTA (DeMamba)**
>
> | Method | Training Data | Recall | F1 | AP |
> | :--- | :---: | :---: | :---: | :---: |
> | **DeMamba** (Reported Avg) | 100% | **93.92** | 90.20 | 97.10 |
> | **MSTFormer** (Ours) | **~1.7%** | 90.19 | **93.50** | **98.50** |
>
> **Q2: The ablation study is incomplete; finer-grained analyses on MSTD (kernels/strides) and CSCL (pairs) are needed.**
>
> **A2:** We have conducted the suggested fine-grained analysis. The results indicate our method is **highly robust** to hyperparameter variations.
>
> *   **Impact of MSTD Settings:** We tested various 3D kernel and stride combinations.
>     | Down-sampling Stride | Kernel Size | ACC | F1 | AP |
>     | :--- | :---: | :---: | :---: | :---: |
>     | $(2,2,2)\rightarrow(4,4,4)\rightarrow(8,8,8)$ | $(3,3,3)$ | 91.31 | 91.12 | 97.08 |
> | $(2,2,2)\rightarrow(4,4,4)\rightarrow(8,8,8)$ | $(5,5,5)$ | 91.59 | 91.36 | 97.16 |
> | $(2,2,2)\rightarrow(4,4,4)\rightarrow(8,8,8)$ | $(7,7,7)$ | 91.44 | 91.39 | 97.07 |
>     | $(1,1,1)\rightarrow(2,2,2)\rightarrow(4,4,4)$ | $(3,3,3)$ | 91.83 | 91.55 | 97.37 |
>     *   *Reason:* The **temporal artifacts** (artificial regularity) in generated videos are pervasive signals observable at multiple scales, making the model insensitive to minor kernel/stride variations.
>
> *   **Impact of CSCL Pair Selection:** Different layer pairing strategies yielded negligible differences.
>     | Strategy (Layer Indices) | ACC | F1 | AP |
>     | :--- | :---: | :---: | :---: |
>     | $(1,2,6,8,14,15)$ | 91.57 | 91.35 | 97.18 |
>     | $(1,2,10,11,14,15)$ | 91.31 | 91.12 | 97.08 |
>     | $(1,2,12,13,14,15)$ | 91.50 | 91.20 | 97.28 |
>     *   *Reason:* The effective fusion of **multi-scale information** is sufficient to contrast semantic-rich deep features with motion-sensitive shallow features, regardless of specific indices.
>
> **Q3: The paper lacks interpretability results (e.g., visualizations) to support the claim about motion discrepancies.**
>
> **A3:** We agree that statistical evidence is crucial. We conducted an analysis of the **temporal variation in optical flow magnitude**. Visualizations are provided in the **Supplementary Material** (Figure X).
> *   **Generated Videos:** Predominantly exhibit **artificial regularity** or uniform stability, lacking natural fluctuations.
> *   **Real Videos:** Characterized by significant **stochasticity (randomness)** and natural abrupt shifts.
> These findings quantitatively verify our premise that AI-generated motion lacks the complexity of real-world footage.
>
> **Q4: Computational costs (FLOPs, parameters, runtime) are not reported to support the "lightweight" claim.**
>
> **A4:** We evaluated MSTFormer using a standard input shape of $(3, 16, 224, 224)$. As shown below, it is significantly more efficient than mainstream backbones (e.g., I3D/TSN range 50-200 GFLOPs).
>
> **Table: Computational Cost**
>
> | Metric | Value | Note |
> | :--- | :---: | :--- |
> | **Parameters** | 10.29 M | Lightweight Model Size |
> | **FLOPs** | 9.96 G | < 20% of typical video backbones |
> | **Inference Time** | 0.22 s | Per video on RTX 4070 SUPER |
>
> **Q5: It is unclear whether re-used videos from GenVideo/GVD in AIDetection are excluded during testing.**
>
> **A5:** We strictly enforced **data isolation**. While AIDetection incorporates subsets from GenVideo/GVD for diversity, the training and testing processes were conducted independently. We guarantee that **no specific video samples used for training were reused or present in the test sets** during cross-dataset evaluations.
>
> We hope these clarifications fully address your concerns. We are happy to engage in further discussion.

---

### Official Review · Reviewer_2RW6 · 2025-11-02

**Soundness:** 2
**Presentation:** 2
**Contribution:** 2
**Rating:** 6
**Confidence:** 4

**Summary:**

This paper tackles the challenge of detecting AI-generated videos, especially under cross-generator OOD generalization. The authors propose a two-part framework called MSTformer.They also introduce a new benchmark dataset AIDetection, containing videos from multiple commercial and closed-source generators as well as real short videos. This dataset aims to reflect realistic, diverse distributions.
Experiments on AIDetection, GVF, and GenVideo show consistent improvements across metrics (ACC, F1, AP). For instance, on AIDetection, MSTformer achieves ACC = 91.31, F1 = 91.12, and AP = 97.08; on GenVideo, ACC = 94.32, F1 = 93.50, and AP = 98.50.
The paper further reports per-generator ACCs, ablations on the two modules, and sensitivity to parameters τ and λ, number of frames, and batch size.

**Strengths:**

1.Clear and targeted motivation.
The authors correctly observe that modern video generators largely eliminate spatial artifacts, making motion dynamics the more reliable cue. The LK flow visualizations in Fig. 1 show that generated videos exhibit stronger foreground–background motion correlation.

2.Method design is consistent with the motivation and technically feasible.
MSTD performs spatiotemporal downsampling via 3D convolution before attention, enlarging the receptive field while preserving local temporal correlation.
CSCL enforces semantic alignment across multiple scales using supervised contrastive loss, preventing ambiguity in single-scale representations.

3.OOD-oriented evaluation setup.
The experiments explicitly test cross-generator generalization under “unknown real sources,” which is highly relevant for real-world robustness.

4.Rich experimental evidence.
Comprehensive comparisons across three datasets, per-generator breakdowns, and one-to-many OOD tests all support the claimed improvements.
Ablation studies confirm that MSTD substantially increases recall (46.56 → 66.81), while CSCL further improves F1 and ACC.

5.Implementation details are sufficiently disclosed.
Frame sampling, optimizer, training setup, and hardware are all clearly stated, aiding reproducibility.

**Weaknesses:**

1.Inconsistent or unclear dataset statistics.
Different sections report inconsistent counts of real/generated samples and generator sources (e.g., 39,298 vs 19,298 real). The authors should reconcile these and provide detailed splits in the appendix.

2.Misaligned evaluation protocol affects external comparability.
The paper explicitly states that, instead of evaluating each generator separately, all test samples are merged.
This change makes results not directly comparable with prior works and could blur whether the gains arise from method improvements or mixed distribution effects.
Suggestion: also report results following the original per-generator protocols (at least in the appendix).

3.Instability on advanced generators (e.g., Sora).
In Table 3, the per-generator ACC on Sora (68.60) is lower than MViTv2-S (71.74), indicating limited robustness to complex temporal motion. The authors should analyze failure cases by motion type, scene dynamics, compression, or frame rate.

4.Limited metrics and fixed-threshold evaluation.
All main results are based on ACC/Precision/Recall/F1/AP computed at a fixed threshold = 0.5.
No ROC-AUC, EER, or calibration analysis is provided.
Suggest adding AUC, EER, and PR-AUC results and discussing calibration or uncertainty under OOD settings.

5.CSCL lacks empirical justification for “cross-scale augmentation.”
Section 4.3 describes semantic consistency qualitatively but omits supporting quantitative evidence (e.g., alignment visualization, mutual information, or ablation comparing single-pair vs. multi-pair combinations).
Suggest including these analyses and reporting contrastive queue statistics in the appendix.


6.Reproducibility and openness.
The release of code and dataset is not guaranteed. The paper should clarify data licensing and provide code/configs or feature files if raw videos cannot be shared.

**Questions:**

.Please confirm the final statistics of AIDetection (real/generated counts, number of generator sources, and split details).
2.Can you reproduce results under the original GVF/GenVideo protocols to enable direct comparison?
3.Why does performance drop on Sora? Is it correlated with scene complexity, motion diversity, or frame rate?
4.For CSCL, what is the pair selection strategy and contrastive queue size? How sensitive is performance to these choices?
5.Could you provide ROC-AUC/EER metrics and discuss calibration or temperature scaling?
6.Have you tested robustness under varying compression levels, frame rates, and resolutions?

---

> ### Author Response · Authors · 2025-12-02
>
> We sincerely thank the reviewer for the constructive feedback and have carefully addressed your questions below.
>
> **Q1: Please confirm the final statistics of AIDetection.**
>
> **A1:** The detailed statistics and organizational structure are comprehensively presented in **Appendix A.2** and **Table 7**. To summarize:
> *   **Generated Videos:** **19,731** samples sourced from **24** different generative paradigms.
> *   **Real Videos:** **19,298** samples collected from **3** distinct sources (including social media platforms).
>
> Table 7 in the paper explicitly details the distribution across these sources, including the specific splits for training and testing sets.
>
> **Q2: Can you reproduce results under original GVF/GenVideo protocols?**
>
> **A2:** While strict pixel-level reproduction is not feasible due to undisclosed details in the original benchmark papers (e.g., exact composition of "Real" videos for specific splits), **MSTFormer demonstrates superior data efficiency**. As shown below, even when trained on a significantly smaller scale (**~1.7%** of the original GenVideo training set), our method achieves results comparable to or better than the reported full-data baselines.
>
> **Table: Performance Comparison with Limited Training Data**
>
> | Method | Recall | F1 | AP | Note |
> | :--- | :---: | :---: | :---: | :--- |
> | **GenVideo (Reported)** | 93.92 | 90.20 | 97.10 | Average over 10 categories |
> | **MSTFormer (Ours)** | 90.19 | **93.50** | **98.50** | Trained with only **~1.7%** data |
>
> This result highlights that MSTFormer does not rely on massive-scale training data to achieve high robustness.
>
> **Q3: Why does performance drop on Sora?**
>
> **A3:** We analyze the slight performance dip ($3.14\%$) on the Sora subset from two perspectives:
> *   **Generalization vs. Specific Overfitting:** While Sora performance dips slightly, we observe significant gains (up to **$35.33\%$**) on other generators like Vidu and Hailuo. This indicates MSTFormer avoids overfitting to specific artifacts, capturing more universal patterns for balanced cross-source generalization.
> *   **Complexity and Motion Dynamics:** We agree with the hypothesis. Sora samples feature **higher frame rates, longer durations, and significantly more complex motion patterns**. These factors result in higher temporal consistency and fewer obvious artifacts, naturally increasing detection difficulty compared to earlier models.
>
> **Q4: CSCL pair selection strategy, queue size, and sensitivity?**
>
> **A4:**
> *   **Pair Selection & Sensitivity:** We select feature sequence pairs from 6 specific stages of the backbone. As shown in the ablation table below, performance fluctuations across different pairing strategies are negligible (F1 ranges narrowly between 91.12% and 91.35%), confirming that MSTFormer is **robust** to specific layer choices provided multi-scale information is fused.
> *   **Queue Size:** The sequence length is determined by the convolutional strides of the down-sampling stages. The channel dimensions are fixed at **96, 192, and 384**, respectively.
>
> **Table: Ablation Study on Pair Selection Strategies**
>
> | Strategy (Layer Indices) | ACC | Precision | Recall | F1 | AP |
> | :--- | :---: | :---: | :---: | :---: | :---: |
> | $(1,2,6,8,14,15)$ | 91.57 | 93.80 | 89.02 | 91.35 | 97.18 |
> | $(1,2,10,11,14,15)$ | 91.31 | 93.10 | 89.23 | 91.12 | 97.08 |
> | $(1,2,12,13,14,15)$ | 91.50 | 94.49 | 88.13 | 91.20 | 97.28 |
>
> **Q5: ROC-AUC/EER metrics and calibration?**
>
> **A5:** We agree that threshold-independent metrics provide valuable perspective on stability. The table below shows that MSTFormer achieves high AUC and low EER across all datasets, indicating excellent separability between real and generated videos without dependence on specific threshold tuning.
>
> **Table: ROC-AUC and EER Metrics**
>
> | Dataset | AUC (%) | EER (%) |
> | :--- | :---: | :---: |
> | **AIDetection** | 96.60 | 8.42 |
> | **GenVideo** | 90.66 | 18.56 |
> | **GVF** | 98.43 | 5.49 |
>
> **Q6: Robustness under varying compression, frame rates, and resolutions?**
>
> **A6:** Yes, we have evaluated robustness across multiple dimensions:
> *   **Frame Rates:** As detailed in **Section 5.2**, the model maintains stable detection capabilities even at low sampling rates (e.g., 8 frames).
> *   **Resolution (Spatial Scaling):** While our main experiments follow the standard protocol (resize & crop to $224 \times 224$), we conducted additional tests with different resize-and-crop strategies. The results below confirm consistent performance, validating robustness to spatial variations.
>
> **Table: Robustness Across Different Resolution Strategies**
>
> | Strategy (Resize $\rightarrow$ Crop) | ACC | Precision | Recall | F1 | AP |
> | :--- | :---: | :---: | :---: | :---: | :---: |
> | $224 \rightarrow 200$ | 91.61 | 94.71 | 88.13 | 91.30 | 97.23 |
> | $256 \rightarrow 224$ | 91.31 | 93.10 | 89.23 | 91.12 | 97.08 |
> | $320 \rightarrow 256$ | 91.83 | 94.73 | 88.58 | 91.55 | 97.37 |
>
> We hope these responses adequately address your concerns.

---

### Meta-Review · Area_Chair_H3jV · 2025-12-24

**Summary:**

The paper addresses the detection of AI-generated videos with a focus on cross-generator OOD generalization. It proposes MSTFormer, which utilizes motion-aware spatiotemporal downsampling, and introduces a new benchmark dataset, AIDetection.

While the reviewers recognized the importance of the OOD setting and the effort to construct a diverse dataset, the consensus leans towards rejection. The primary grounds for this decision are limited technical novelty and experimental shortcomings. Reviewers bqV7 pointed out that the architecture is largely an incremental adaptation of existing works (like RecoNet/MViT-v2) without significant structural innovation. Furthermore, critical concerns regarding the fairness of benchmarking remain: specifically, the misalignment of evaluation protocols preventing direct comparison with SOTA (2RW6, yYLf), and the notable performance drop on advanced generators like Sora (2RW6), which questions the method's longevity.

**Reviewer Concerns:**

**Addressed Concerns:**

1. Missing Metrics: The authors successfully provided the AUC and EER metrics requested by 2RW6, showing reasonable threshold-independent performance.

2. Computational Cost: The concern raised by yYLf regarding the "lightweight" claim was addressed by providing FLOPs and parameter counts.

3. Statistical Clarifications: The inconsistencies in dataset statistics pointed out by 2RW6 were clarified in the rebuttal.

**Outstanding Concerns (The basis for Rejection):**

1. Limited Novelty: This is the most significant outstanding issue. bqV7 correctly identified that the core modules (feature pooling, spatiotemporal attention) are standard components found in prior works like RecoNet and MViT-v2. The authors' rebuttal, claiming novelty lies in the temporal application of these blocks, was viewed as an engineering adaptation rather than a methodological breakthrough.

2. Comparison Fairness & Baselines: yYLf noted the omission of recent SOTA methods. While the authors added comparisons against reported statistics, the lack of reproduction on the exact same splits makes these comparisons scientifically loose. Similarly, 2RW6's concern about the merged evaluation protocol (vs. per-generator) remains a hindrance to fair benchmarking against prior art.

3. Robustness on SOTA Generators: 2RW6 highlighted the performance drop on Sora (approx. 68%). The authors' explanation that Sora has fewer artifacts actually underscores a limitation: the method relies on artifacts present in older generators and may not generalize well to the high-fidelity video synthesis era.

4. Dataset Significance: bqV7 noted that the new dataset is significantly smaller (39k) than existing benchmarks (GenVideo), limiting its potential impact as a standalone contribution.

**Reviewer Scores:**

- 2RW6: (Original: Marginally above acceptance) -> Borderline Reject. While the reviewer initially gave a positive score, the rebuttal confirmed that the method struggles with advanced generators (Sora) and relies on a non-standard evaluation protocol, which weakens the validity of the results.

- yYLf: (Original: Marginally below acceptance) -> Reject. The reviewer's core concern about missing strict, fair comparisons with SOTA (e.g., DeMamba on the exact same splits) was not fully resolved, as the authors relied on loose comparisons with reported stats.

- bqV7: (Original: Marginally below acceptance) -> Reject. The reviewer's primary concern regarding the incremental nature of the technical contribution (similarity to RecoNet) remains unchanged.

---

### Decision · Program_Chairs · 2026-01-26

Reject